# Computer-Aided Planning in Orthognathic Surgery: A Comparative Study with the Establishment of Burstone Analysis-Derived 3D Norms

**DOI:** 10.3390/jcm8122106

**Published:** 2019-12-02

**Authors:** Cheng-Ting Ho, Rafael Denadai, Hsin-Chih Lai, Lun-Jou Lo, Hsiu-Hsia Lin

**Affiliations:** 1Division of Craniofacial Orthodontics, Department of Dentistry, Chang Gung Memorial Hospital, Taoyuan City 333, Taiwan; ma2589@gmail.com (C.-T.H.); sam985tw@gmail.com (H.-C.L.); 2Department of Plastic and Reconstructive Surgery, Craniofacial Research Center, Chang Gung Memorial Hospital, Chang Gung University, Taoyuan City 333, Taiwan; denadai.rafael@hotmail.com (R.D.); lunjoulo@cgmh.org.tw (L.-J.L.); 3Image Lab and Craniofacial Research Center, Chang Gung Memorial Hospital, Taoyuan City 333, Taiwan

**Keywords:** three-dimensional norms, two-dimensional norms, Burstone analysis, orthognathic surgery, simulation, computer-aided simulation

## Abstract

Three-dimensional (3D) computer-aided simulation has revolutionized orthognathic surgery treatment, but scarce 3D cephalometric norms have been defined to date. The purposes of this study were to (1) establish a normative database of 3D Burstone cephalometric measurements for adult male and female Chinese in Taiwan, (2) compare this 3D norm dataset with the two-dimensional (2D) Burstone norms from Caucasian and Singaporean Chinese populations, and (3) apply these 3D norms to assess the outcome of a computer-aided simulation of orthognathic surgery. Three-dimensional Burstone cephalometric analysis was performed on 3D digital craniofacial image models generated from cone-beam computed tomography datasets of 60 adult Taiwanese Chinese individuals with normal occlusion and balanced facial profile. Three-dimensional Burstone analysis was performed on 3D image datasets from patients with skeletal Class III pattern (*n* = 30) with prior computer-aided simulation. Three-dimensional Burstone cephalometric measurements showed that Taiwanese Chinese males had significantly (*p* < 0.05) larger anterior and posterior facial heights, maxillary length, and mandibular ramus height than females, with no significant (*p* > 0.05) difference for facial soft-tissue parameters. The 3D norm dataset revealed Taiwanese Chinese-specific facial characteristics, with Taiwanese presenting (*p* < 0.05) a more convex profile, protrusive maxillary apical bases, protruding mandible, protruding upper and lower lips, and a shorter maxillary length and lower facial height than Caucasians. Taiwanese had significantly (*p* < 0.05) larger maxillary projection, vertical height ratio, lower face throat angle, nasolabial angle, and upper lip protrusion than Singaporean Chinese. No significant (*p* > 0.05) difference was observed between 3D norms and computer-aided simulation-derived 3D patient images for horizontal skeletal, vertical skeletal, and dental measurements, with the exception of two dental parameters (*p* < 0.05). This study contributes to literature by providing gender- and ethnic-specific 3D Burstone cephalometric norms, which can assist in the multidisciplinary-based delivery of orthodontic surgical care for Taiwanese Chinese individuals worldwide, including orthodontic management, computer-assisted simulation, and outcome assessment.

## 1. Introduction

Three-dimensional (3D) computer-assisted simulation has revolutionized orthognathic surgery treatment [1,2]. An accurate diagnosis of actual dento-skeletofacial deformity is a key step in successful 3D digital image-based surgical simulation [3,4]. Normative data and cephalometric analysis are thus routinely adopted during 3D computer-assisted orthodontic and surgical treatment workflows [5,6,7,8,9,10].

In 1978, Burstone developed the “Cephalometrics for Orthognathic Surgery” [11,12]—a detailed analytical method which is widely accepted and used in the fields of orthodontics and orthognathic surgery [13,14,15,16,17,18,19,20,21,22,23,24]. This two-dimensional (2D) image-based cephalometric method includes clinically useful facial bone (linear and angular measurements in horizontal and vertical directions) and soft-tissue (facial form and lip position) analysis [11,12]. Moreover, the Burstone’s analysis presents a particular parameter—that is, the Sn-Pog’ line—which was previously described as relevant for attractiveness-related perceptions in the Chinese population [25]. However, the 2D imaging modality presents limitations, such as a lack of volumetric information, enlargement, distortion, the overlap of important anatomical structures, and restrictions of anatomical landmark identification, affecting accurate diagnosis and planning [26,27,28,29]. Moreover, these static images fail to provide complete information for 3D skeletofacial structures, especially in patients with facial asymmetry of the midline and contour [29,30,31].

Over the past years, cone-beam computed tomography (CBCT) scans have been adopted to overcome 2D image-related drawbacks [1,2]. CBCT scans provide a precise visualization of anatomical structures and pathologic processes due to their high-quality 3D image acquisition and reconstruction parameters, including the accurate and reliable positioning of the acquired virtual head to achieve 3D coordinates of cephalometric landmarks and measurements [26,27,28,29,30]. It was also shown that 2D radiographic versus 3D CBCT-based cephalometric measurements present statistically significant and clinically relevant differences [26,27,28,29,30]. Therefore, traditional 2D cephalometric norms are not ideal for 3D CBCT-based cephalometric analysis.

Clinicians should consider the existing gender- and ethnic-related differences to achieve or preserve the particular desired facial features of adult patients under orthodontic and orthognathic surgery treatments. However, limited gender- and ethnic-specific 3D cephalometric normative values have been reported to date, with most of the existing data not addressing elements such as the overall facial features [5,6,7,8,9,10]. In addition, the absence of Burstone analysis-derived 3D norms results in the need for multidisciplinary teams to adopt the 2D Burstone norms for 3D computer-assisted clinical practice and research. These aspects would limit the real application of the full potential benefits and advantages of 3D digital imaging modality in orthodontics and orthognathic surgery, including the diagnosis, planning, execution, and validation of treatment.

The primary purpose of this study was to establish a normative database of 3D Burstone cephalometric measurements for adult male and female Chinese in Taiwan. Secondary purposes were to compare male and female data, compare this 3D Burstone normative dataset with the 2D Burstone norms, and apply these 3D Burstone norms to assess the outcome of the computer-aided simulation of orthognathic surgery.

## 2. Methods

This study recruited 60 normal volunteers (30 male and 30 female, aged 20–30 years, with a mean age of 24.5 years; Figure 1) based on incidental contacts from members of the Chang Gung University and Chang Gung Memorial Hospital, Taiwan. The inclusion criteria were (1) Taiwanese Chinese ethnicity; (2) balanced facial features; (3) proper incisor overbite and overjet (0–2 mm) with no or mild crowding; (4) Class I skeletal relation (ANB = 3–5 degrees); (5) willing to receive CBCT scans; and (6) sufficient 3D image quality. The exclusion criteria were (1) contraindication for CBCT examination; (2) congenital or acquired dento-skeletofacial deformity or orbital canting; (3) any facial surgical intervention; and (4) any previous orthodontic treatment.

This study was performed with the approval of the Institutional Review Board (Chang Gung Medical Foundation = 201600686A3). All included individuals provided informed consent for participation.

### 2.1. Three-Dimensional Image Acquisition and Processing

A standard craniofacial CBCT scan was performed for each individual using an i-CAT 3D Dental Imaging System (Imaging Sciences International, Hatfield, PA, USA) with the following parameters: 120 kVp, 0.4 × 0.4 × 0.4 mm voxel size, 40 s scan time, and 22 × 16 cm field of view. The patient’s head was positioned with the Frankfort horizontal plane parallel to the ground. Throughout the scan, patients were instructed not to swallow, to keep their mouth closed, and to maintain a centric occlusion bite. Images were stored in Digital Imaging and Communications in Medicine format and rendered into 3D volumetric images using Simplant O&O software package (Materialize Dental, Leuven, Belgium) [29,30,31,32].

### 2.2. Three-Dimensional Burstone Cephalometric Analysis

To set a normative database of 3D Burstone cephalometric measurements for Taiwanese Chinese, a total of 27 linear and nine angular measurements were generated using Simplant software. For this, all craniofacial bone and soft-tissue anatomical landmarks and reference planes (22, 14, and 8 parameters, respectively) were standardized based on previous descriptions (Table 1 and Table 2, Figure 2 and Figure 3) [11,12,33,34,35]. Two Burstone analysis-related anatomical landmarks (i.e., PtM and Ar) [11,12] were adapted; the anterior base length (S-N) replaced the cranial base length (Ar-PtM-N) and the Co-Go replaced Ar-Go for ramus length measurement. Due to the relevance of cheek prominence for overall aesthetic balance in Asians [25,36,37,38], a further parameter—that is, cheek mass (CK) (Figure 4)—was added to the Burstone soft-tissue analysis.

After the orientation of the 3D digital models at a standardized position, all anatomical landmarks and reference planes were marked and created by a single operator. Twenty randomly selected 3D models were marked in a two-week interval by the same operator, with intra-operator errors being analyzed by calculating the Euclidean distance between the first and second anatomical landmark coordinates.

### 2.3. Three-Dimensional Versus 2D Norms

To appraise the ethnic element of normative cephalometric data (Figure 1), the original Burstone’s 2D norms derived from a Caucasian population—that is, European–Americans [11,12]—were compared with the 3D Taiwanese Chinese norms created in this study. The 2D Burstone soft-tissue norms derived from the Singaporean Chinese population were also adopted for comparative analysis [24]. Only the parameters with complete information in previous reports [11,12,24] were adopted for analysis.

### 2.4. Outcome of Computer-Assisted Simulation

To appraise the applicability of the created 3D Taiwanese Chinese norms, a previously published [30] sample of orthognathic surgery-treated patients was enrolled. Preoperative 3D image datasets of 30 Taiwanese Chinese patients with a skeletal Class III pattern (Figure 1) who had undergone computer-guided simulation by the same multidisciplinary team were retrieved from the electronic records of the Imaging Laboratory, Chang Gung Craniofacial Research Center.

Full descriptions of the 3D computer-aided simulation adopted in this center were previously detailed [29,30,31,32]. After the segmentation of Le Fort I maxillary and bilateral sagittal split osteotomies, the conventional orthodontic 2D plan (Figure 5) was transferred into the virtual scenario; this planning and transferring considered the 2D Taiwanese Chinese cephalometric norms. Through a collaborative teamwork approach between orthodontist and surgeon, further adjustments were performed in midline, roll, pitch, and yaw directions, as well as genioplasty. These patient-specific skeletofacial adjustments were performed before surgical intervention [30], and no bone movement was implemented for the current study. The final 3D image datasets adopted for the transferring of the simulation to actual surgery were adopted as a basis to perform 3D cephalometric measurements using the Burstone analysis method. The 3D cephalometric dataset resulting from this analysis was compared with the 3D norms generated in this study. A 3D digital image dataset was also randomly selected to be adopted in a computer-aided simulation using the Burstone analysis-derived 3D norms.

### 2.5. Statistical Analysis

In the descriptive analysis, data were presented as means ± standard deviations. Independent *t* tests were adopted for comparative analysis. The Pearson correlation test was adopted for reliability analysis, with a higher Pearson correlation coefficient (*r* ranging from 0 to 1) indicating a higher reliability. Two-sided values of *p* < 0.05 were considered statistically significant. All analyses were performed using SPSS Version 15.0 (IBM Corp., Armonk, NY, USA).

## 3. Results

### 3.1. Reliability Analysis

Higher Pearson correlation coefficients (*r* = 0.88–1.00, all *p* < 0.05) revealed excellent intra-operator reliability for all anatomical landmark identifications (Table 3).

### 3.2. Three-Dimensional Facial Bone Norms

Taiwanese Chinese males had a longer (*p* < 0.05) anterior cranial base (S-N) than females, with a difference of 5 mm. For the vertical skeletal and dental parameters, the anterior upper facial height (N-ANS), posterior upper facial height (N-PNS), and posterior dental height (6-MP) were found to be 3 mm longer (all *p* < 0.05) in males than in females. For maxilla/mandible parameters, the maxillary length (ANS-PNS) and mandibular ramus height (Co-Go) were longer (all *p* < 0.05) in males than in females, with differences of 5 and 8 mm, respectively. No significant difference was observed in the horizontal skeletal relationship and dental inclination measurements (Table 4).

### 3.3. Three-Dimensional Facial Soft-Tissue Norms

No significant difference was observed between Taiwanese Chinese males and females for the facial soft-tissue measurements (Table 5).

### 3.4. Three-Dimensional Versus 2D Norms

Considering the facial bone parameters, Taiwanese Chinese males had significantly (*p* < 0.05) larger (N-A, Go-Pg, OP-HP, and U1-NF) and smaller (ANS-Gn, U1-NF, L1-MP, PNS-ANS, and B-Pg) values than Caucasian males. Taiwanese Chinese females had significantly (*p* < 0.05) larger (N-A, N-B, N-Pg, N-ANS, ANS-Gn, Go-Pg, and OP-HP) and smaller (PNS-ANS, B-Pg, and A-B) values than Caucasian females (Table 6). Considering the facial soft-tissue parameters, Taiwanese Chinese individuals had significantly (*p* < 0.05) larger mandibular projection (G-Pg’) and upper and lower lip protrusions (Ls to Sn-Pg’ and Li to Sn-Pg’) than Caucasian individuals (Table 7).

Considering the facial soft-tissue parameters, Taiwanese Chinese individuals had significantly (*p* < 0.05) larger values (G-Sn, G-Sn/Sn-Me, Sn-Gn’-C, Cm-Sn-Ls, and Ls to Sn-Pg’) than Singaporean Chinese individuals (Table 8).

### 3.5. Outcome of Computer-Assisted Simulation

No significant difference was observed in the comparison between 3D Burstone analysis from 3D patient images after computer-guided simulation and 3D Taiwanese Chinese norms for all horizontal skeletal, vertical skeletal and dental, and dental measurements, with the exception of two dental parameters with larger (OP-HP angle) and smaller (L1-MP angle) values in the 3D patient images than in the 3D norms (Table 9). A practical example of 3D computer-guided surgical simulation using the 3D Taiwanese Chinese norms is presented in Figure 6 and Table 10 and Table 11.

## 4. Discussion

A growing number of studies have revealed that 3D computer-aided orthognathic surgery planning and execution outperforms the traditional 2D planning method for many meaningful parameters such as the correction of midline deviation, mandible ramus asymmetry, occlusal plane canting, and chin position, as well as in terms of cost-effectiveness and patient-reported outcomes [3,29,39,40,41]. To complete the 3D digital image-assisted orthognathic surgery pathway, it is necessary to determine gender- and ethnic-specific 3D cephalometric norms for both the facial bone and soft-tissue components. Three-dimensional cephalometric norms of the Chinese population were previously reported [6,7,8], but orthognathic surgery-specific data and facial regional variations were not fully addressed.

By using the advanced biomedical engineering software-based 3D digital image measurement methodology, 3D Burstone cephalometries were performed on Taiwanese Chinese adults with class I occlusion and a well-balanced facial profile. Overall, the appraisal of the current 3D Taiwanese Chinese norms for males and females reveals some similarities and dissimilarities compared with previously published 3D cephalometric datasets [5,6,7,8,9]. This highlights the value of gender- and ethnic-specific normative data, which are of paramount importance for the precise analysis and planning of orthodontics and orthognathic surgery treatments [5,6,7,8,9].

Taiwanese Chinese males had larger bone-based linear and angular values than females (Table 4). Males had a longer cranial base (S-N) than females, with no difference in sagittal skeletal relationships and the angle of facial convexity (N-A-Pg). Korean and Turkish populations [5,9] also had similar results in the gender-related comparison. Taiwanese Chinese individuals had larger angles of facial convexity (6.2 degrees, Table 7) than Korean and Turkish patients (4.9 and 2.9 degrees, respectively) [5,9], indicating a more protruding facial appearance in the Taiwanese Chinese population. Taiwanese Chinese males also had larger anterior and posterior upper facial heights than females, which is coincident with the Korean population [5]. Taiwanese Chinese males had a longer dental height in the lower molars to MP (L6-MP) than females, demonstrating a longer male lower face.

Similar to the Korean population [5], Taiwanese Chinese males had a significantly longer maxillary length than females, which corresponds to findings of the anterior cranial base [42]. Also, similar to Koreans [5], Taiwanese Chinese males had a longer ramus length (Co-Go) than females. However, Taiwanese Chinese males had a longer maxillary length and ramus height (53.4 and 65.8 mm, respectively) than Korean males (47.9 and 61.2 mm, respectively) [5].

No significant difference was observed between Taiwanese Chinese males and females for facial soft-tissue measurements (Table 5). Males had larger facial convexity angles, upper lip protrusions, and vertical lip–chin ratios than females, but not significantly. Hong Kong Chinese males also had larger lower lip protrusion and depth of the labiomental fold values than females [7]. Moreover, the gender-related differences of lower lip protrusion and labiomental fold values were smaller in Taiwanese Chinese adults (0.7 and 0.3 mm, respectively) than in Hong Kong Chinese adults (1.6 and 0.9 mm, respectively) [7].

In this study, cheek prominence (Figure 4) was measured as an additional parameter to the Burstone analysis. Patients with skeletal Class III patterns—a prevalent deformity in Asian populations—frequently present with maxillary hypoplasia, inadequate support of the infraorbital areas, and paranasal depression in the mid-face [43,44]. On average, the CK point was positioned 2.1 mm anterior to the cornea level. This 3D cheek prominence data can be applied during the decision regarding the amount of Le Fort I maxillary movement required to achieve a balanced cheek contour.

To further contemplate the ethnic element in 3D cephalometry, the 3D Burstone norms developed in this study were compared with the 2D Burstone norms [11,12,24]. Taiwanese Chinese males exhibited a more protruded maxillary apical base, steeper occlusal plane, and more proclined upper incisors, but shorter lower facial height and maxillary length and less chin prominence than Caucasian males (Table 6). Taiwanese Chinese females also showed a more protruded maxilla and mandible and steeper occlusal plane, but a shorter upper facial height and shorter facial height and maxillary length than the Caucasian females. For facial soft-tissue parameters, Taiwanese Chinese individuals had larger mandibular projection and upper and lower lip protrusions than Caucasians (Table 7). These findings are coincident with previous reports displaying the Chinese facial morphology as presenting a more protrusive jawbone base, dentoalveolus, and lips, as well as a more acute nasolabial angle than that of Caucasians [22,45,46]. Overall, these results assist in the clarification of differences between Chinese and European–American populations [22,45,46,47,48]. It also reinforces that, when Chinese adults are to be assessed, the Caucasian norms cannot be employed as benchmark values of diagnosis, planning, or outcome assessment. Moreover, Taiwanese Chinese individuals had larger maxillary projection, vertical height ratio, lower face throat angle, nasolabial angle, and upper lip protrusion than Singaporean Chinese individuals (Table 8), underlining interethnic deviation within Chinese populations [24,46,47].

Taiwanese Chinese individuals also had a significantly larger mandibular body length (Go-Pg) than Caucasians. However, an inverse result was previously described, with Chinese adults presenting with smaller midfaces and shorter mandibles compared with Caucasians [48]. The Pg point-related differences between 2D (3D structure projecting onto a single plane with distortion of the correct location) and 3D (three spatial planes) imaging modalities should be considered as potential explanatory factors for the dissimilarities founded in our current 3D-based findings and the previous 2D-based study [48]. As the 2D linear distances were previously [48] measured based on projected images rather than the true spatial distance, the current 3D linear measurement presented larger values. These findings emphasize the requirement of 3D virtual imaging to appraise the skeletofacial morphology and proportion with no interference from 2D-related anatomical distortions [26,27,28,29]. The constant proliferation of centers adopting the 3D digital technology as the first-line imaging modality may drastically reduce the 2D-related limitations—that is, size and shape distortion, superimposition, and misrepresentation of anatomical structures—in the future.

In this study, we also embraced the newly developed 3D Burstone cephalometric norms to measure the outcome of computer-assisted surgical simulation. Three-dimensional digital image datasets from a previously described cohort [30] were used for 3D cephalometric analysis using the Burstone method. These patients had a steeper occlusal plane (OP-HP) and smaller lower incisor-to-MP angle (L1-MP) than the normal Taiwanese individuals (Table 7), which reflects their deformity (skeletal Class III pattern) and our therapeutic approach [30]. The pitch clockwise rotation of the occlusal plane (posterior impaction), designed to achieve a more convex profile, has resulted in an increase of MP angle and shortening of the posterior facial height. Moreover, in the surgery-first approach routinely used in this center [49,50,51], the incisal inclination has been corrected postoperatively by using the advantage of the regional acceleratory phenomenon [51].

Despite these dental differences, no significant differences were observed between the simulation-derived 3D patient images and the 3D norms for the horizontal and vertical skeletal and vertical dental measurement parameters (Table 7). This implies that the adoption of 2D Taiwanese Chinese norms as an initial guide of planning plus patient-specific 3D virtual adjustments resulted in cephalometric values within the normative data for these profile-related parameters. This outcome reflects the 3D image-guided clinical judgment of orthodontist and surgeon professionals as they interactively judged the skeletal framework changes after the final surgical occlusion setup and transfer of the 2D planning into a virtual scenario. They also contemplated the bone framework morphology and its relationship to the soft-tissue envelope. Using 3D simulation, the complete judgment of frontal, profile, and basal views allows the translational and rotational movements of the maxilla and the proximal and distal segments of the mandibular ramus to be accurately tailored to the need of each patient under treatment [3,4,29,30,31,32].

While these findings could be interpreted as a satisfactory outcome of 3D computer-assisted surgical simulation, only the profile view was considered for analysis, and this should be considered when interpreting the results. The expressive advantages of 3D computer-assisted simulation in a frontal view—for example, facial symmetry—were previously demonstrated [3,29,30], but with no use of 3D cephalometric norms for simulation or outcome measurement. Future studies should assess the role of the adoption of 3D cephalometric norms as reference guides for simulation as well as the inclusion of additional views for outcome analysis. We hypothesized that using the 3D cephalometric norms during simulation would reduce the requirement of some types of bone adjustments due to the elimination of 2D-related anatomical distortions.

This study presents further limitations. Excellent intra-operator reliability was evidenced for all anatomical landmark identifications, but no inter-examiner reliability was provided. While 3D normative data was defined for male and female Taiwanese Chinese individuals, other nontested factors (i.e., body height) were not considered, deserving further investigation. Moreover, the studies of soft tissue cephalometric norms in Caucasian and Singaporean Chinese populations did not perform gender-stratified analysis [12,24], limiting a direct comparison with the current 3D Taiwanese Chinese data. The Bustone cephalometric method used in this study is a quantitative technique commonly employed for orthognathic surgical planning and outcome assessment [11,12,13,14,15,16,17,18,19,20,21,22,23,24]. However, other methods exist which use 3D digital image-based measurements to enhance the arsenal of strategies and possibilities of clinicians. As the newly created 3D norms were compared with the 2D norms, potential differences in the adopted image methods should be considered when interpreting these results. Moreover, expected interethnic variation within Chinese populations, sample size disparities, and methodological dissimilarities (2D radiographic, CBCT, and stereophotogrammetric-derived images) may explain, at least partially, the differences between our current and previous findings [7,24,47], which is worthy of further investigation. As new 3D cephalometric databases are being created across different ethnicities, further comparative studies may be performed to expand our current findings.

The application of the 3D Burstone cephalometric norms exemplified their potential use in clinical practice (with a patient as a practical example, as shown in Figure 6 and Table 10 and Table 11) and research (such as the outcome of a previously published cohort in Table 9), but it was restricted to patients with skeletal Class III deformity who were managed by a particular orthognathic surgery approach and principle for 3D virtual surgery [29,30,31,32]. The use of other groups is also encouraged to assess their orthognathic surgery cohorts to expand upon our findings by enrolling other patients (e.g., skeletal Class II deformity and cleft skeletofacial deformity) who are managed with different orthodontic–surgical approaches (single-jaw surgery or two-splint technique) and principles for 3D simulation. Future studies may also consider this current 3D norms dataset as a reference basis to expand the outcome-based research of orthognathic surgery—for example, the correlation between 3D Burstone cephalometric norms and the perception of facial appearance and aesthetic—which could enhance patient-centered care and the shared decision-making process by providing better counseling to patients and family members.

Despite these drawbacks, this study presents the first gender- and ethnic-specific 3D Burstone cephalometric norms. This current 3D digital image-derived normative dataset can assist professionals (dentists, orthodontists, oral surgeons, maxillofacial surgeons, ear, nose, and throat surgeons, plastic surgeons, and head and neck surgeons) working in multidisciplinary teams to delivery orthognathic surgery care for Taiwanese Chinese individuals distributed worldwide, including preoperative diagnosis, orthodontic management, 3D digital image-assisted simulation, and outcome assessment. To achieve a comprehensive and accurate full-face measurement for future 3D-based diagnosis, treatment planning, and outcome assessment, it is important that the 3D Burstone cephalometric norms are also defined for other ethnicities.

## 5. Conclusions

This study contributes to literature by providing gender- and ethnic-specific 3D Burstone cephalometric norms.

## Figures and Tables

**Figure 1 jcm-08-02106-f001:**
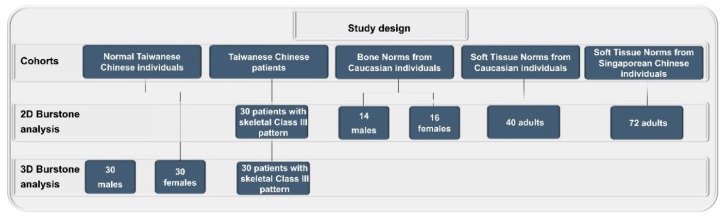
Flowchart of the different cohorts and types of cephalometric analysis enrolled in this study. For further information, please, refer to the Section 2.2, Section 2.3 and Section 2.4.

**Figure 2 jcm-08-02106-f002:**
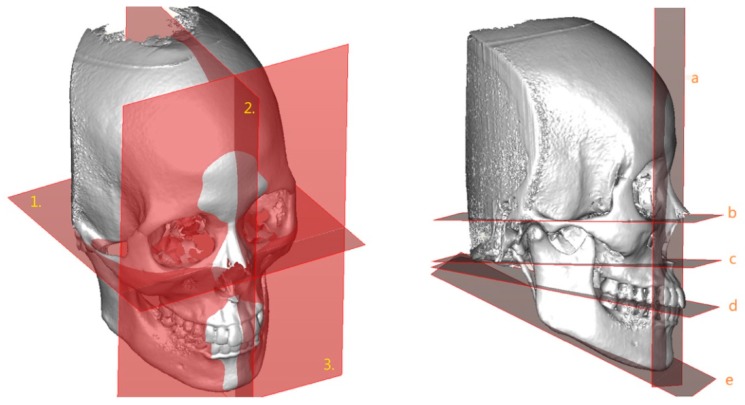
Three-dimensional skeletofacial model displaying the reference planes used for cephalometric analysis: (1) X, horizontal plane; (2) Y, midsagittal plane; (3) Z, vertical plane; (a) N plane; (b) FH plane; (c) palatal plane; (d) occlusal plane; and (e) mandibular plane.

**Figure 3 jcm-08-02106-f003:**
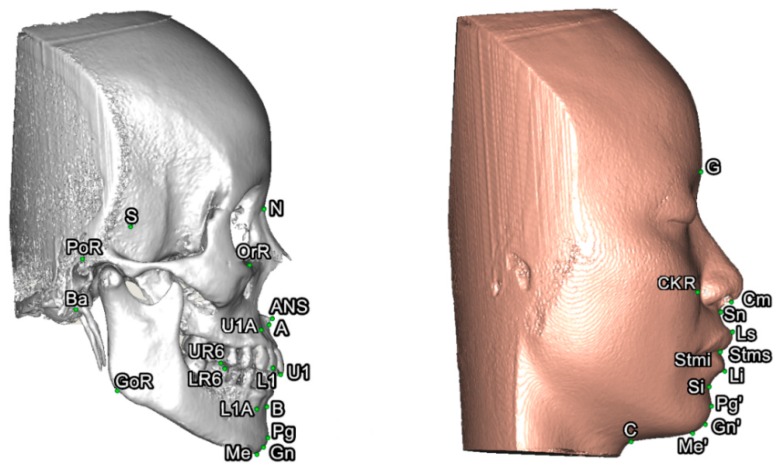
Three-dimensional skeletofacial model displaying the bone and soft-tissue landmarks adopted for cephalometric analysis. For definitions, please, refer to Table 1.

**Figure 4 jcm-08-02106-f004:**
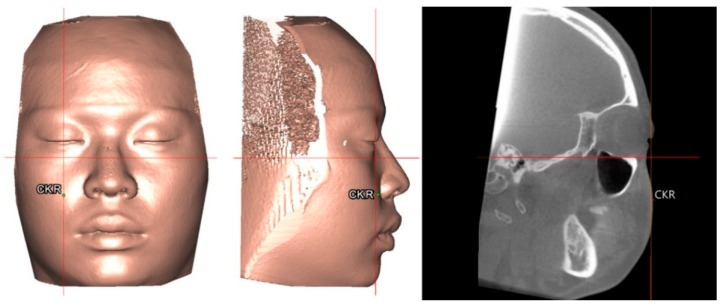
Location of cheek mass (CK) for cheek prominence measurement. To represent the most projected point on the cheek contour, the CK point was located on the most convex point on the mid-pupillary plane (vertical line) below the infraorbital area. The horizontal distance between CK and mid-pupillary vertical line was measured. For definitions, please, refer to Table 1 and Table 2.

**Figure 5 jcm-08-02106-f005:**
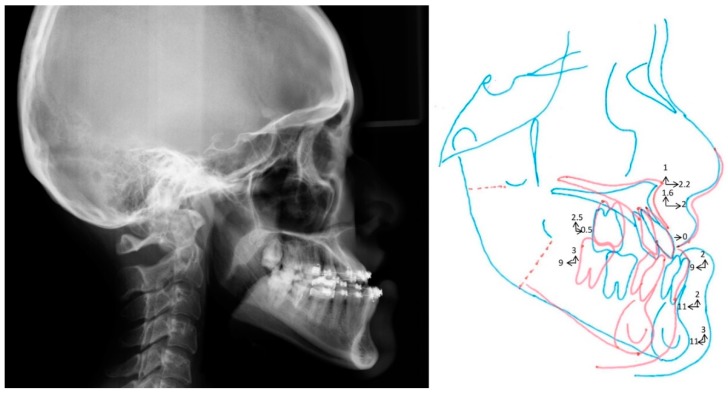
Practical example of the two-dimensional (2D) planning based on 2D Burstone norms for orthognathic surgery treatment using the single-splint, two-jaw surgery technique. Blue and red colors symbolize the initial bone framework and final maxilla–mandibular repositioning, respectively. Arrows represent the direction and amount of bone segment movements.

**Figure 6 jcm-08-02106-f006:**
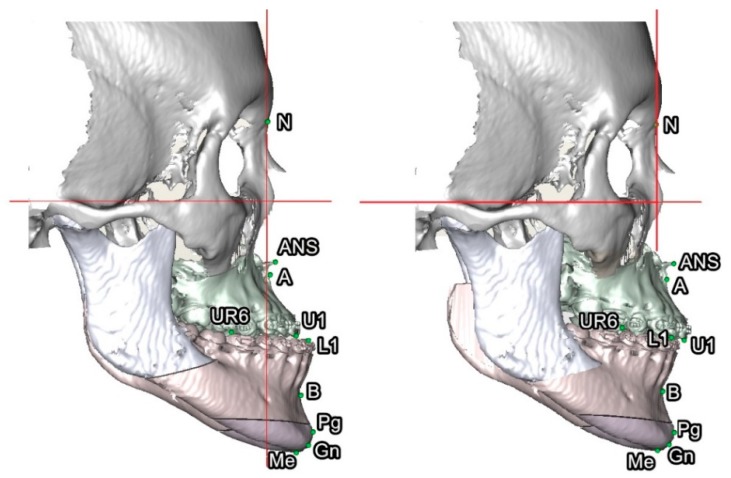
Practical example of the 3D surgical simulation based on 3D norms for orthognathic surgery treatment using the single-splint, two-jaw surgery technique. Skeletal Class III deformity (left) before and (right) after computer-aided simulation. After setting the planned final occlusion setup by mobilization of osteotomized distal mandible segment in direction to the fixed maxilla, the maxillomandibular complex (MMC; composed by Le Fort I and distal mandible segments) was moved as a single unit in 6 degrees of freedom (including translational and rotational directions) to meet the normal position of point A to N vertical line, angle of convexity, Pog, vertical ratio, and symmetry. Pitch clockwise rotation of MMC was needed to fit the best facial convexity and obtain a better smile arc. The final modifications were decided by the surgeon and orthodontist according to the clinical evaluation of the soft tissue facial profile. For definitions, please, refer to Table 1 and Table 2.

**Table 1 jcm-08-02106-t001:** Definition of the anatomical landmarks adopted for three-dimensional (3D) cephalometric analysis.

Landmarks	Abbreviations	Definitions
**Skeletal Landmark**		
**Cranium**		
Orbitale	Or	The most inferior point of each infra-orbital rim
Porion	Po	The most superior point of each external acoustic meatus
Sella turcica	S	The center of the sella turcica on the midsagittal plane
Nasion	N	The junction between the nasal and frontonasal sutures
Basion	Ba	The most anterior point of the foramen magnum
**Maxilla**		
Anterior nasal spine	ANS	The most anterior midpoint of the anterior nasal spine of the maxilla
Posterior nasal spine	PNS	The most posterior midpoint of the posterior nasal spine of the palatine bone
A point	A	The point of maximum concavity in the mid-line of the alveolar process of the maxilla
Posterior maxillary point	PMP	The point of maximum concavity of the posterior border of the palatine bone in the horizontal plane at both sides
**Mandible**		
B point	B	The point of maximum concavity in the mid-line of the alveolar process of the mandible
Pogonion	Pog	The most anterior midpoint of the chin on the outline of the mandibular symphysis
Menton	Me	The most inferior midpoint of the chin on the outline of the mandibular symphysis
Gnathion	Gn	The most anterior and inferior point on the anterior margin of symphysis on the sagittal plane
Gonion	Go	Dropping a perpendicular from the intersection point of the tangent lines to the posterior margin of the mandibular vertical ramus and inferior margin of the mandibular body or horizontal ramus
Condylion	Co	The most postero-superior point of each mandibular condyle in the sagittal plane
**Dentoalveolar Landmarks**		
U1 incisal tip	U1	The midpoint between the crowns of the maxillary central incisors tip
U1 incisal apex	U1A	The incisor apex of the upper central incisor
L1 incisal tip	L1	The incisal tip of the crown of lower central incisor
L1 incisor apex	L1A	The midpoint between the crowns of the mandibular central incisors tip
U6 cusp	UR6, UL6	The most inferior point of the mesial cusp of the crown of each first upper molar in the profile plane
L6 cusp	LR6, LL6	The most superior point of the mesial cusp of the crown of each first lower molar in the profile plane
**Soft tissue Landmarks**		
Glabella	G	The most anterior midpoint on the front-to-orbital soft tissue contour
Columella	Cm	The point on each columella crest, level with the top of the corresponding nostril
Subnasale	Sn	The midpoint on the nasolabial soft tissue contour between the columella crest and the upper lip
Labiale superius	Ls	The midpoint of the vermilion line of the upper lip
Stomion	Stm (Stms, Stmi)	The midpoint of the horizontal labial fissure (Stms, upper lip; Stmi, lower lip)
Labiale inferius	Li	The midpoint of the vermilion line of the lower lip
Sublabiale	Si	The most posterior midpoint on the labiomental soft tissue contour that defines the border between the lower lip and the chin
Soft tissue pogonion	Pg’	The most anterior point of the soft tissue chin in the centerline
Soft tissue gnathion	Gn’	The most inferior point of the soft tissue chin in the centerline
Cervical point	C	The junction of the submental, the submandibular regions and the neck in the midline
Menton’	Me’	The most inferior midpoint of the chin on the outline of the soft tissue over mandible
Cheek mass	CK	The most convex point under infraorbital area relative to the perpendicular line from midpoint of upper eyelid to FH plane
Cornea	CL, CR	The most anterior point of the cornea

L, left side; R, right side.

**Table 2 jcm-08-02106-t002:** Definition of the reference planes adopted for 3D cephalometric analysis.

Planes	Definition
Frankfort horizontal plane (FH)	A plane through landmarks Orbitale (Or) on both sides and the midpoint of Porion (Po) of both sides
Midsagittal plane (MS)	A plane formed by basion (Ba), nasion (N), and perpendicular to FH plan
Palatal plane (PP)	A plane through landmarks ANS and PMP on both sides
Occlusal plane (OP)	A plane through the mean of landmarks upper incisor tip and lower incisor tip on both sides (U1Tip & L1 Tip), through the means of landmarks upper and lower molar buccal cusp of both sides
Mandibular plane (MP)	A plane through landmark Menton (Me) and Gonion (Go) of both sides
Anterior facial plane (N-vert)	A plane through landmark of Nasion (N) and perpendicular to FH plane and MS plane
Soft tissue anterior facial plane (G-vert)	A plane through landmark of Glabella (G) and perpendicular to FH plane and MS plane
Mid-pupillary plane	A plane through Cornea point (C) and perpendicular to FH plane and coronal plane

**Table 3 jcm-08-02106-t003:** The accuracy of the identification of anatomical landmarks in the 3D coordinate system.

Parameters	Mean Difference (mm)	SD	Median	Q1	Q3	*r*	*p*-Value
Basion (Ba)	0.32	0.13	0.31	0.23	0.41	1.00	0.004
Nasion (N)	0.30	0.19	0.27	0.15	0.48	0.93	0.009
Sella turcica (S)	0.33	0.26	0.65	0.58	0.71	0.89	0.001
Porion (Po)	0.42	0.52	0.71	0.52	1.03	0.88	0.008
Orbitale (Or)	0.48	0.53	0.73	0.40	1.14	0.96	0.003
Posterior maxillary point (PMP)	0.43	0.15	0.43	0.30	0.54	0.99	0.008
Anterior nasal spine (ANS)	0.43	0.21	0.39	0.29	0.59	0.99	0.009
Posterior nasal spine (PNS)	0.50	0.219	0.47	0.33	0.63	0.92	0.004
A point (A)	0.35	0.54	0.52	0.24	0.64	0.91	0.007
B point (B)	0.36	0.18	0.41	0.19	0.48	0.88	0.004
Pogonion (Pog)	0.38	0.37	0.22	0.15	0.63	0.93	0.006
Gnathion (Gn)	0.49	0.22	0.35	0.24	0.53	0.96	0.003
Menton (Me)	0.40	0.48	0.37	0.25	0.87	0.90	0.006
Gonion (Go)	0.49	0.61	0.77	0.45	1.09	0.89	0.006
Condylion (Co)	0.46	0.62	0.79	0.36	1.37	0.90	0.003
U1 incisal tip (U1T)	0.34	0.12	0.34	0.28	0.40	0.98	0.003
U1 incisal apex (U1A)	0.46	0.40	0.48	0.36	0.64	0.91	0.006
L1 incisal tip (L1T)	0.35	0.35	0.32	0.17	0.38	0.99	0.001
L1 incisor apex (L1A)	0.32	0.09	0.35	0.26	0.37	0.99	0.001
U6 cusp (UR6C, UL6C)	0.38	0.17	0.32	0.29	0.43	0.99	0.007
L6 cusp (LR6C, LL6C)	0.48	0.27	0.48	0.34	0.64	0.97	0.006
Glabella (G)	0.43	0.30	0.49	0.33	0.69	0.99	0.009
Columella (Cm)	0.43	0.35	0.35	0.23	0.38	0.99	0.003
Subnasale (Sn)	0.44	0.23	0.44	0.26	0.66	0.97	0.009
Labiale superius (Ls)	0.39	0.32	0.42	0.29	0.49	0.99	0.007
Stomion (Stm)	0.28	0.15	0.28	0.19	0.32	0.99	0.005
Labiale inferius (Li)	0.36	0.24	0.34	0.14	0.56	0.90	0.003
Soft tissue pogonion (Pog’)	0.39	0.37	0.22	0.15	0.63	0.91	0.006
Soft tissue gnathion (Gn’)	0.45	0.37	0.34	0.30	0.44	0.99	0.001
Cervical point (C’)	0.40	0.51	0.21	0.04	0.71	0.96	0.008
Mean ± SD	0.40 ± 0.06						

*r*, Pearson correlation coefficient; SD, standard deviation; Q1, lower quantile; Q3, upper quantile.

**Table 4 jcm-08-02106-t004:** Comparison of Burstone analysis-based 3D bone cephalometric norms between Taiwanese Chinese males and females.

Parameters	Male (*n* = 30)		Female (*n* = 30)		*p*-Value
	Mean	SD	Mean	SD	
**Cranial Base**					
S-N mm	67.941	1.911	62.563	3.191	0.006
**Horizontal (Skeletal)**					
N-A-Pg (angle) °	6.506	2.987	5.850	2.945	0.610
N-A (//HP) mm	2.552	1.588	2.124	1.263	0.493
N-B (//HP) mm	−1.791	4.434	−2.517	3.991	0.691
N-Pg (//HP) mm	−1.062	4.521	−1.731	4.438	0.731
**Vertical (Skeletal, Dental)**					
N-ANS (PHP) mm	55.457	4.299	51.052	1.293	0.007
ANS-Gn (PHP) mm	65.093	5.043	63.744	2.803	0.449
PNS-N (PHP) mm	54.195	2.143	50.569	2.097	0.001
MP-HP (angle) °	23.939	4.134	25.321	1.820	0.307
U1-NF (PNF) mm	28.240	3.599	28.101	1.148	0.905
U6-NF (PNF) mm	24.818	2.754	24.490	2.777	0.784
L6-MP (PMP) mm	35.418	2.865	32.368	2.388	0.014
L1-MP (PMP) mm	42.056	2.710	41.430	1.972	0.543
**Maxilla, Mandible**					
PNS-ANS (//HP) mm	53.431	2.726	48.240	3.200	0.001
Co-Go (linear) mm	65.866	6.482	57.665	3.064	0.002
Go-Pg (linear) mm	92.061	4.793	88.908	4.582	0.130
B-Pg (//MP) mm	5.415	0.985	5.361	3.461	0.740
**Dental**					
OP upper-HP (angle) °	8.915	2.286	9.139	2.444	0.827
U1-NF (angle) °	115.272	5.951	112.090	7.175	0.272
L1-MP (angle) °	96.685	9.558	92.932	2.906	0.236
A-B (//OP) mm	−1.742	1.177	−2.456	1.497	0.228

° degree (angle); mm, millimeters; SD, standard deviation; //HP, //MP, and //OP indicate measurements parallel to the horizontal plane, mandibular plane, and occlusal plane, respectively; PHP, PNF, and PMP indicate measurements perpendicular to the horizontal plane, nasal floor, and mandibular plane, respectively. For the definition of parameters, please refer to Table 1 and Table 2.

**Table 5 jcm-08-02106-t005:** Comparison of Burstone analysis-based 3D soft-tissue cephalometric norms between Taiwanese Chinese males and females.

Parameters	Male (*n* = 30)		Female (*n* = 30)		*p*-Value
	Mean	SD	Mean	SD	
**Facial Form**					
Facial convexity angle: G′-Sn′-Pg′	12.245	3.415	10.572	2.642	0.214
Maxillary projection: G-Sn (//HP)	6.920	2.168	6.489	2.428	0.665
Mandibular projection: G-Pg′ (//HP)	1.445	3.579	2.843	3.362	0.378
Vertical height ratio: G-Sn/Sn-Me (PHP)	1.114	0.095	1.126	0.102	0.775
Lower face throat angle: Sn-Gn′-C	100.818	7.319	100.501	3.375	0.899
Cheek mass (cheek contour): CK	2.105	1.028	2.186	1.312	0.257
**Lip Position and Form**					
Nasolabial angle: Cm-Sn-Ls	98.651	8.694	99.796	8.279	0.755
Upper lip protrusion: Ls to (Sn-Pg′)	6.426	1.846	5.472	1.717	0.224
Lower lip protrusion: Li to (Sn-Pg′)	4.475	1.455	3.771	1.847	0.333
Mentolabial sulcus: Si to (Li-Pg′)	3.923	0.832	3.672	0.763	0.469
Vertical lip chin ratio: Sn-Stm/Stm-Me′ (PHP)	0.521	0.053	0.488	0.043	0.131
Maxillary incisor exposure: Stm-1	1.182	1.079	1.455	1.128	0.569
Interlabial gap: Stms-Stmi (PHP)	1.273	0.647	1.182	0.603	0.737

SD, standard deviation; //HP indicates measurements parallel to the horizontal plane; PHP indicates measurements perpendicular to the horizontal plane. For the definition of parameters, please refer to Table 1 and Table 2.

**Table 6 jcm-08-02106-t006:** Comparison between 3D Taiwanese Chinese norms and 2D Burstone’s Caucasian norms for facial bone parameters.

Parameters	Taiwanese Male (*n* = 30)		Caucasian Male (*n* = 14)		*p*-Value	Taiwanese Female (*n* = 30)		Caucasian Female (*n* = 16)		*p*-Value
	Mean	SD	Mean	SD		Mean	SD	Mean	SD	
**Horizontal (Skeletal)**										
N-A-Pg (angle)	6.506	2.987	3.9	6.4	0.161	5.850	2.945	2.6	5.1	0.056
N-A (//HP)	2.552	1.588	0.0	3.7	0.022	2.124	1.263	−2.0	3.7	0.001
N-B (//HP)	−1.791	4.434	−5.3	6.7	0.086	−2.517	3.991	−6.9	4.3	0.007
N-Pg (//HP)	−1.062	4.521	−4.3	8.5	0.198	−1.731	4.438	−6.5	5.1	0.011
**Vertical (Skeletal, Dental)**										
N-ANS (PHP)	55.457	4.299	54.7	3.2	0.576	51.052	1.293	50.0	2.4	0.016
ANS-Gn (PHP)	65.093	5.043	68.6	3.8	0.034	63.744	2.803	61.3	3.3	0.001
PNS-N (PHP)	54.195	2.143	53.9	1.7	0.671	50.569	2.097	50.6	2.2	0.968
MP-HP (angle)	23.939	4.134	23.0	5.9	0.601	25.321	1.820	24.2	5.0	0.433
U1-NF (PNF)	28.240	3.599	30.5	2.1	0.047	28.101	1.148	27.5	1.7	0.260
U6-NF (PNF)	24.818	2.754	26.2	2.0	0.119	24.490	2.777	23.0	1.3	0.076
L6-MP (PMP)	35.418	2.865	35.8	2.6	0.696	32.368	2.388	32.1	1.9	0.738
L1-MP (PMP)	42.056	2.710	45.0	2.1	0.002	41.430	1.972	40.8	1.8	0.371
**Maxilla, Mandible**										
PNS-ANS (//HP)	53.431	2.726	57.7	2.5	<0.001	48.240	3.200	52.6	3.5	0.001
Go-Pg (linear)	92.061	4.793	83.7	4.6	<0.001	88.908	4.582	74.3	5.8	<0.001
B-Pg (//MP)	5.415	0.985	8.9	1.7	<0.001	5.361	3.461	7.2	1.9	0.084
**Dental**										
OP upper-HP (angle)	8.915	2.286	6.2	5.1	0.007	9.139	2.444	7.1	2.5	0.032
U1-NF (angle)	115.272	5.951	111.1	4.7	0.040	112.090	7.175	112.5	5.3	0.864
L1-MP (angle)	96.685	9.558	95.9	5.2	0.776	92.932	2.906	95.9	5.7	0.101
A-B (//OP)	−1.742	1.177	−1.1	2.0	0.285	−2.456	1.497	−0.4	2.5	0.017

SD, standard deviation; //HP, //MP, and //OP indicate measurements parallel to the horizontal plane, mandibular plane, and occlusal plane, respectively; PHP, PNF, and PMP indicate measurements perpendicular to the horizontal plane, nasal floor, mandibular plane, respectively. For the definition of parameters, please refer to Table 1 and Table 2.

**Table 7 jcm-08-02106-t007:** Comparison between 3D Taiwanese Chinese norms and 2D Burstone’s Caucasian norms for facial soft tissue parameters.

Parameters	Taiwanese (*n* = 60) *		Caucasian (*n* = 40) **		*p*-Value
	Mean	SD	Mean	SD	
**Facial Form**					
Facial convexity angle: G′-Sn′-Pg′	11.409	3.100	12	4.0	0.482
Maxillary projection: G-Sn (//HP)	6.705	2.257	6.0	3.0	0.263
Mandibular projection: G-Pg′ (//HP)	2.144	3.626	0	3.0	0.005
Vertical height ratio: G-Sn/Sn-Me (PHP)	1.130	0.088	1.0	–	–
Lower face throat angle: Sn-Gn′-C	100.660	5.881	100	7.0	0.660
**Lip Position and Form**					
Nasolabial angle: Cm-Sn-Ls	99.224	8.305	102	8.0	0.134
Upper lip protrusion: Ls to (Sn-Pg′)	5.949	1.807	3.0	1.0	<0.001
Lower lip protrusion: Li to (Sn-Pg′)	4.122	1.662	2.0	1.0	<0.001
Mentolabial sulcus: Si to (Li-Pg′)	3.797	0.789	4.0	2.0	0.595
Vertical lip chin ratio: Sn-Stm/Stm-Me′(PHP)	0.505	0.050	0.5	–	–
Maxillary incisor exposure: Stm-1	1.319	1.044	2.0	2.0	0.088
Interlabial gap: Stms-Stmi (PHP)	1.227	0.571	2.0	2.0	–

SD, standard deviation; //HP indicates measurements parallel to the horizontal plane; PHP indicates measurements perpendicular to the horizontal plane; – indicates data not available; * Combined data (30 males and 30 females); ** Combined data (20 males and 20 females). For the definition of parameters, please refer to Table 1 and Table 2.

**Table 8 jcm-08-02106-t008:** Comparison between 3D Taiwanese Chinese norms and 2D Burstone’s Chinese norms from Singapore for facial soft tissue parameters.

Parameters	Taiwanese (*n* = 60) *		Singaporean (*n* = 72) **		*p*-Value
	Mean	SD	Mean	SD	
**Facial Form**					
Facial convexity angle: G′-Sn′-Pg′	11.409	3.100	10.5	3.5	0.164
Maxillary projection: G-Sn (//HP)	6.705	2.257	2.5	3	<0.001
Mandibular projection: G-Pg′ (//HP)	2.144	3.626	–	–	–
Vertical height ratio: G-Sn/Sn-Me (PHP)	1.130	0.088	1.0	0.1	<0.001
Lower face throat angle: Sn-Gn′-C	100.660	5.881	96	4	<0.001
**Lip Position and Form**					
Nasolabial angle: Cm-Sn-Ls	99.224	8.305	95	3	0.001
Upper lip protrusion: Ls to (Sn-Pg′)	5.949	1.807	7	1.5	0.002
Lower lip protrusion: Li to (Sn-Pg′)	4.122	1.662	–	–	–
Mentolabial sulcus: Si to (Li-Pg′)	3.797	0.789	3.5	2	0.336
Vertical lip chin ratio: Sn-Stm/Stm-Me′ (PHP)	0.505	0.050	0.5	–	–
Maxillary incisor exposure: Stm-1	1.319	1.044	1.5	1.5	0.481
Interlabial gap: Stms-Stmi (PHP)	1.227	0.571	1	1	0.167

SD, standard deviation; //HP indicates measurements parallel to the horizontal plane; PHP indicates measurements perpendicular to the horizontal plane; – indicates data not available; * Combined data (30 males and 30 females); ** Combined data (36 males and 36 females). For the definition of parameters, please refer to Table 1 and Table 2.

**Table 9 jcm-08-02106-t009:** Comparison between 3D Burstone analysis of computer-guided simulation models and 3D Taiwanese Chinese norms.

Parameters	3D Simulation (*n* = 30)		3D Norms (*n* = 60) *		*p*-Value
	Mean	SD	Mean	SD	
**Horizontal (Skeletal)**					
N-A-Pg angle	4.750	2.873	6.178	2.914	0.111
N-A (//HP)	1.114	3.284	2.338	1.417	0.119
N-B (//HP)	−2.773	4.618	−2.154	4.133	0.673
N-Pg (//HP)	−1.764	4.066	−1.396	4.385	0.716
**Vertical (Skeletal, Dental)**					
N-ANS/ANS-Gn	0.831	0.066	0.826	0.097	0.614
MP-HP (angle)	25.001	2.637	24.636	3.156	0.446
U1-NF/L1-MP	0.675	0.062	0.677	0.046	0.919
**Dental**					
OP upper-HP (angle)	12.341	2.337	9.027	2.312	<0.001
U1-NF (angle)	112.818	4.992	113.68	6.483	0.491
L1-MP (angle)	88.318	2.495	94.809	7.156	<0.001

SD, standard deviation; * Combined data (Taiwanese Chinese males and females); //HP indicates measurements parallel to the horizontal plane. For the definition of parameters, please, refer to Table 1 and Table 2.

**Table 10 jcm-08-02106-t010:** Data from computer-guided surgical simulation in a patient with skeletal Class III deformity using the 3D norms.

Parameters	Initial	Surgical Simulation
**Horizontal (Skeletal)**		
N-A-Pg	−14.23	2.69
Nvert-A	1.43	4.09
Nvert-B	12.63	2.97
Nvert-Pg	17.37	7.14
**Vertical (Skeletal, Dental)**		
N-ANS	48.95	48.19
ANS-Gn	66.89	65.62
N-ANS/ANS-Gn (ratio)	0.73	0.73
U1-NF	27.00	26.97
L1-MP	38.24	38.13
U1-NF/L1-MP	0.71	0.71
**Dental**		
OP-FH	2.81	9.49
UR6-NP	47.68	44.63
UL6-NP	49.04	46.36

**Table 11 jcm-08-02106-t011:** Data from computer-guided surgical simulation (maxillary advancement and mandible setback) in a patient with skeletal Class III deformity using the 3D norms.

Parameters	Treatment Plan in 3 Dimension (mm)		
	*X* Axis (Left, Right)	*Y* Axis (Anterior, Posterior)	*Z* Axis (Up−Down)
N	0	0	0
ANS	0	2.75	0.76
A point	0	2.37	1.03
B point	0	−10.35	2.11
Pog	0	−10.92	1.96
Gn	0	−10.66	2.03
Me	0	−10.93	2.14
U1 mid	0	0	0.01
L1 mid	0	−9.00	2.01
U6R	0	0.40	2.94
U6L	0	0.20	2.70
Right (+) Left (−)		Anterior (+) Posterior (−)	Up (+) Down (−)

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
