# Peer review of "Computer-Aided Planning in Orthognathic Surgery: A Comparative Study with the Establishment of Burstone Analysis-Derived 3D Norms"

_jcm, 2019, doi:10.3390/jcm8122106_

Round 1

Reviewer 1 Report

The study established a normative database of 3D Burstone cephalometric measurements, compared the 2D with the 3D measurements from Caucasians and Singapore Chinese population and finally, compared the 3D norms with the outcome of orthognathic surgical patients.

The authors have performed an excellent study with an interesting research question and meticulous investigation. I have only 1 minor comment to the methods used in the repeated measurements analysis. The variance of the repeated measurements is not mentioned, and should be provided along with the mean. And since the repeatability measurements are performed on only 10 patients, the data is usually nor considered normally distributed, and therefore it may be better to provide the median and range of the measurement instead of the mean and SD.

On a personal preference towards repeated measurements analysis, using the Pearson’s correlation coefficient or the intraclass correlation coefficient (ICC) only provides information on the correlation between the 2 measurements and not the absolute difference between the measurements. Personally, I prefer to use the methods of Bland and Altman with the 95 % Limit of agreement which refers to the difference between the measurements and thus the reliability of the measurements, which is the purpose of performing repeated measurements. However, this is just a personal note and not needed in for publication of this study.

Overall, an impressive study.

Reviewer 2 Report

I think this investigation looks so organized and well completed. This would be a great work, which will contribute to the advances of this field in 3D era. 

But , I wonder why you chose the Burstone's analysis. There are so many cephalometric analysis. I think you'd better describe the reason why you have chosen the Burstone's analysis. 

And the comparisonal figures or illustrations between the 2D ceph and 3D ceph will be a help for the understanding of the readers. 

Reviewer 3 Report

This manuscript entitled ‘Computer-aided planning in orthognathic surgery: a comparative study with the establishment of Burstone analysis-derived 3D norms’ investigated linear and angular measurements from 60 subjects who had balanced profile and normal occlusion and compared the measurements to those obtained from Caucasian and Singaporean Chinese. In addition, the authors performed three-dimensional (3D) surgical simulation for patients having Class III malocclusion and compared the simulation results to the measurements obtained from the 60 ‘normal’ subjects. Although the authors did a lot of effort to perform this research, the reviewer thinks it is not appropriate to be published in the Journal of Clinical Medicine because of a lack of originality, wrong study design, low data reliability, and inappropriate statistical methods.

First, the reviewer was not able to find out the originality of this study or clinical importance. As the authors mentioned, 3D cephalometric norms were previously reported for the Chinese population. Although the authors stated that orthognathic surgery-specific data and facial regional variations were not fully addressed, this study did not cover the two things but performed surgical simulations by transferring two-dimensional (2D) surgical plans to 3D images.

Second, the study design needs to be revised. The aims of this study were to establish normative data of 3D measurements. However, all the measurements except the cheek contour were based on 2D measurements, which are exactly the same as conventional lateral cephalometry. Therefore, it seemed that the authors wanted to know the measurements from 3D data for the normal group would be the same as those from 2D data. It means that the authors should verify first 3D data can be compared directly to the 2D data because the magnification ratio would be different. Please note that cone-beam computed tomography (CBCT) and lateral cephalogram obtain images by a totally different way, which makes different magnification. Furthermore, the authors showed that the surgical simulation results were matched with the normative data. The reviewer wonders what the purpose was to compare the two data and if the simulations have any clinical impact.

Regarding the data reliability, please clarify how the authors obtained the raw data for Caucasian and Singaporean Chinese. It is also required that the authors should verify inter-examiner reliability. Also, the reviewer was not able to understand the inclusion criteria of normal subjects. The inclusion criteria such as balanced facial features cannot guarantee the subjects have normal or representative craniofacial structures. In particular, the linear measurements would differ according to subjects’ sex or body height. The comparison between Male and Female groups was not included in the purpose of this study.

Last, please explain why the authors compared the Male and Female groups, which were not the purpose of this study. We all know that men have larger dimensions than women. Please present how many subjects were used for Caucasian Male, Female, and Singaporean; if the other population data showed normal distribution; and why the Caucasian group was divided into male and female only in Table 6.

Overall, this study has a novelty only for the cheek prominence in Taiwanese Chinese, which was very limited and barely added academic and clinical findings.

The followings are minor corrections and suggestions:  

In the Abstract, the purposes of this study were not matched with the conclusions. First, please clarify this study was performed to establish three-dimensional (3D) Taiwanese Chinese norms.

Please present the unit of each variable in Table 4.

The following landmarks and reference planes need to be explained in more detail:

For APMP, there are two definitions CK, CL, and CR should be explained separately Regarding the Mid-pupillary plane, please confirm that the landmark of the cornea is one.

Round 2

Reviewer 3 Report

The authors have made appropriate changes to the manuscript. 

Thank you for the corrections.